# Mitochondria-Targeted Catalase Does Not Suppress Development of Cellular Senescence during Aging

**DOI:** 10.3390/biomedicines12020414

**Published:** 2024-02-10

**Authors:** Bronwyn A. Mogck, Samantha T. Jezak, Christopher D. Wiley

**Affiliations:** 1Jean Mayer USDA Human Nutrition Research on Aging, Tufts University, Boston, MA 02111, USA; 2SENS Research Foundation, Mountain View, CA 94041, USA; 3Friedman School of Nutrition Science & Policy, Tufts University, Boston, MA 02111, USA; 4Department of Medicine, School of Medicine, Tufts University, Boston, MA 02111, USA

**Keywords:** cellular senescence, reactive oxygen specific, aging, catalase, mCAT, SASP

## Abstract

Cellular senescence is a complex stress response marked by stable proliferative arrest and the secretion of biologically active molecules collectively known as the senescence-associated secretory phenotype (SASP). Mitochondria-derived reactive oxygen species (ROS) have been implicated in aging and age-related processes, including senescence. Stressors that increase ROS levels promote both senescence and the SASP, while reducing mitochondrial ROS or mitochondria themselves can prevent senescence or the SASP. Mitochondrially targeted catalase (mCAT), a transgene that reduces mitochondrial levels of ROS, has been shown to extend the lifespan of murine models and protect against the age-related loss of mitochondrial function. However, it remains unclear whether mCAT can prevent senescence or the SASP. In this study, we investigated the impact of mCAT on senescence in cultured cells and aged mice in order to discover if the lifespan-extending activity of mCAT might be due to the reduction in senescent cells or the SASP. Contrary to expectations, we observed that mCAT does not reduce markers of senescence or the SASP in cultured cells. Moreover, mCAT does not prevent the accumulation of senescent cells or the development of the SASP in adipose tissue from aged mice. These results suggest that mitochondrial ROS may not always play a causal role in the development of senescence during natural aging and underscore the need for a nuanced understanding of the intricate relationship between mitochondrial ROS and cellular senescence.

## 1. Introduction

The loss of mitochondrial function is a potentially important driver of aging and can limit the life and health span of mammals [1,2,3]. One aspect of this loss is an increase in mitochondrial reactive oxygen species (ROS) as these organelles are a major site for ROS generation. Murine knockouts of antioxidant enzymes such as superoxide dismutases 1 and 2 (SOD1 and SOD2) and catalase (CAT) are short-lived [4,5,6,7], indicating that cellular antioxidant defenses are required for normal life and health spans. Furthermore, increasing antioxidant proteins or treatment with antioxidants can extend the life span of invertebrate models [8,9,10]. Despite these data, the overexpression of most antioxidant enzymes does not extend the life span of mice [11,12], suggesting that antioxidant defenses in these animals are already sufficient for geroprotection under unstressed conditions. A notable exception to this occurs in the case of a mitochondrially targeted catalase (mCAT) transgene [13]. In this model, catalase—which converts hydrogen peroxide into O_2_ and water—specifically targets mitochondria, providing these organelles with an added layer of protection from a common source of ROS-mediated damage. These mice live 10–20% longer than wild-type (WT) mice and are protected from the age-related loss of mitochondrial function [13], but it remains unclear if mCAT can attenuate the development of other aspects of aging, such as cellular senescence.

Cellular senescence is a stress or damage response characterized by a proliferative growth arrest accompanied by the release of various cytokines, chemokines, growth factors, proteases, oxylipins, and other signaling molecules collectively known as the senescence-associated secretory phenotype (SASP) [14,15,16,17]. Senescent cells have been linked to a number of age-related diseases and can limit both life and health spans, as the elimination of these cells protects against the development of several age-related pathologies [18,19,20,21,22]. Importantly, mitochondrial dysfunction and ROS can drive cellular senescence in culture, as well as in the skin and adipose tissue of mice [23,24].

We previously demonstrated that mitochondrial dysfunction can result in a senescent phenotype that lacks multiple proinflammatory features found in the SASP. This mitochondrial-dysfunction-associated senescence (MiDAS) occurs in response to alterations in the cytosolic NAD+/NADH ratio, regardless of ROS status [23], indicating that mitochondrial dysfunction may drive senescence independent of ROS production; however, other models suggest that mitochondrial ROS may drive nuclear DNA damage or downstream signaling events that result in senescence and the SASP [24,25,26]. It is therefore unclear if reducing mitochondrial ROS is effective in reducing the burden of senescent cells or the SASP during natural aging.

Here, we show that transgenic mCAT has no effect on senescent phenotypes in cultured human fibroblasts. Furthermore, gonadal adipose tissue from aged WT and mCAT mice shows increases in many markers of senescence both at 17 and after 25 months, but mCAT has no discernable effect on these markers. Together, these data support a model in which mitochondrial ROS are not universally required for senescence or the SASP during natural aging.

## 2. Materials and Methods

**Cell culture.** IMR-90 human fibroblasts were cultured in Dulbecco’s Modified Eagle Medium (DMEM) supplemented with 10% fetal bovine serum as well as penicillin and streptomycin. Cells were cultured at 37 °C in 3% O_2_ and 5% CO_2_ and used between 25 and 40 population doublings. All cultures were mycoplasma-free.

**Lentiviral transduction of mCAT.** The *mCAT* transgene (a kind gift from Peter Rabinovitch, and described in [13]) was cloned into the 670-1 lentiviral vector and packaged as previously described [27]. IMR-90 fibroblasts were transduced with a lentivirus containing either an *mCAT* transcript or an empty vector, then selected for successful transduction via 2 days of culture in 2 μg/mL puromycin before being using in this study.

**Induction of senescence.** Senescence was induced in cultured cells through irradiation [SEN(IR)] and mitochondrial DNA (mtDNA) depletion, as described previously [23]. Briefly, cells were irradiated with 10 Gy of ionizing radiation (IR), and control cells were mock-irradiated. Mitochondrial DNA was depleted for MiDAS cells by culturing cells in the presence of 100 ng/mL ethidium bromide and 50 μg/mL uridine for 14 days. Rho cells were generated by culturing cells for 2 months in the presence of 100 ng/mL ethidium bromide, 1 mM of sodium pyruvate, and 50 μg/mL uridine. The confirmation of mtDNA depletion was performed on DNA extracts by quantitative PCR for mtDNA/nDNA ratios, normalized to untreated cells (Appendix A). Primers can be found in Appendix A.

**Animal models.** Aged WT control and *mCAT* mice were in the C57Bl/6 background and were described previously [13], and gonadal adipose tissue from these mice was a kind gift from Peter Rabinovitch (University of Washington). One cohort represented a set of mice that were homogenously euthanized at 17 months of age, before the onset of mortality in this study. The second cohort represented mice 24–30 months of age with approximately equivalent median ages, as described in Appendix A of [13]. Young control (4 mo) C57Bl/6 mice were bred and maintained at the Buck Institute until 4 months after birth, at which point the animals were euthanized and their tissues were harvested. Procedures for young (4 mo) mice were approved by the Buck Institute Institutional Animal Care and Use Committee.

**IL-6 ELISA.** Conditioned media were generated via 24 h of culture in serum-free media, followed by collection and centrifugation. Cells were then trypsinized and counted for normalization purposes. The levels of IL-6 in the conditioned media were measured by bead-based ELISA (AlphaLISA, Perkin-Elmer, Waltham, MA, USA) and performed according to the manufacturer’s protocol. In short, AlphaLISA Anti-Analyte Acceptor beads and DIG-labeled Anti hIL-6 antibody were added to the samples, followed by a 1 h incubation period and the addition of Anti-Digoxigenin Fab Fragment Donor Beads. After the addition of the donor beads, samples were incubated for 30 min in the dark, and then fluorescence was detected at 615 nm. All results were normalized to the cell number.

**Quantitative real-time PCR.** RNA was isolated from both cells and mouse gonadal fat using commercial kits (Bioline for cells, Qiagen for adipose tissue). Isolated RNA was reverse-transcribed using random hexamer primers and analyzed by qPCR using the Universal Prove Library system (Roche, Basel, Switzerland), with normalization to actin. Primer sequences and probe numbers are given in Appendix A.

**Mitochondrial ROS.** 500,000 cells were cultured in the presence of 2 µM MitoSOX Red in Phenol Red-free DMEM or 2 µM DCFDA and incubated at 3% O_2_ and 5% CO_2_. After 45 min of incubation, cells were trypsinized, resuspended in PBS, and analyzed on a Cytek Guava flow cytometer. Output files were gated and analyzed for red (MitoSOX)- or green (DCFDA)-channel fluorescence intensity using FlowJo v.7.6.5 software. Histograms were generated using Floreada, version: SIMD.

**EdU Incorporation.** Cells were cultured in the presence of 10 μM EdU in growth medium for 24 h, then fixed for 10 min in 4% buffered formalin and washed. Cells were permeabilized for 30 min in 0.5% Triton X-100 and then treated according to the manufacturer’s instructions (Life Technologies Cat #C10337, Carlsbad, CA, USA).

**Immunofluorescence/Immunohistochemistry.** Immunofluorescence and immunohistochemistry were performed as described previously [23,28]. Briefly, cultured cells were washed in PBS and fixed in 10% neutral buffered formalin for 10 min. Cells were washed three times in ice-cold PBS and permeabilized in 0.5% Triton X-100 for 30 min, followed by three additional washes. Cells were blocked in 10% normal buffered goat serum for 30 min, followed by the addition of 53BP1 antibody (Novus) in goat serum for 1 h. Cells were washed three times in PBS, followed by the addition of Alexa-594-conjugated goat–anti-rabbit secondary antibody for 30 min. Cells were washed and mounted using Vectastain mounting medium with DAPI. Stained cells were imaged on slides using a fluorescence microscope and quantified with Cell Profiler using the Speckle Counting Pipeline.

Gonadal adipose tissues from mice were fixed in 10% buffered formalin, embedded in paraffin, and then sectioned into 5–7 μm slices, which were incubated with an HMGB1 antibody (Abcam ab18256, Cambridge, UK) overnight. Immunohistochemistry was performed on the sections using the Vectastain Elite ABC-HRP KIT (Vector Labs, Newark, CA, USA) according to the instructions of the manufacturer. The quantitation of senescent cells was performed using 10 non-overlapping fields and by dividing either beta-galactosidase-positive cells or HMGB1-positive nuclei by the total number of nuclei.

**Senescence-associated beta galactosidase.** Staining for senescence-associated beta-galactosidase was accomplished as previously described [2] using a commercial kit (Biovision-Abcam, Milpitas, CA, USA). Cells were cultured, washed in PBS, and fixed for 10 min. Cells were then washed 3 times with ice-cold PBS, followed by treatment with staining solution, and then incubated at 37 °C overnight. Wells were imaged with a microscope and analyzed for the number of beta-galactosidase-positive cells using the blue channel in ImageJ2 version: 2.14.0/1.54f.

For tissue staining, 50 μg gonadal adipose tissue samples were fixed in 500 μL of fixation solution for 15 min, followed by 3 washes using ice cold PBS. Tissues were then placed in 500 μL of staining solution and incubated at 37 °C with observation every 3 h until color (blue) developed. Stained tissues were photographed together and analyzed for beta-galactosidase intensity using the blue channel in ImageJ2 version: 2.14.0/1.54f.

## 3. Results

### 3.1. Transgenic mCAT Does Not Antagonize Senescence Phenotypes in Human Fibroblasts

To determine if mitochondrial hydrogen peroxide is a driver of senescence phenotypes, we transduced IMR-90 fibroblasts, a commonly used cell type for the mechanistic study of senescence, with a lentivirus containing a mitochondrially targeted catalase (*mCAT*) construct, or an empty vector. Following selection, senescence was induced in cells by either 10 Gy of ionizing radiation [SEN(IR)] or the depletion of mitochondrial DNA by serial culture in the presence of ethidium bromide until senescent (MiDAS). We chose these two inducers as they have been shown to drive senescence and elevate mitochondrial ROS levels [23,24,26]. Vector- and *mCAT*-transduced cells were irradiated with the same dose of IR or cultured in EtBr for the same amount of time, and mitochondrial DNA levels were confirmed by quantitative PCR (Appendix A). Despite reductions in MitoSOX and DCFDA fluorescence in mCAT-transduced calls (Figure 1A,B and Appendix A), no changes in growth kinetics were observed between WT and mCAT cells cultured under identical conditions. Ten days after IR or mock irradiation—or 14 days after starting EtBr treatment—cells were analyzed for senescence markers, including increased *p21^WAF1^* and decreased *LMNB1* RNA levels [29,30] (Figure 1C,D), senescence-associated beta-galactosidase [31] (Figure 1E,F), and EdU incorporation (Figure 1G). No changes were observed for any of these parameters, indicating that mCAT did not affect most major cellular senescent phenotypes.

Since mitochondrial ROS have been implicated as drivers of the SASP [24,25], we also sought to determine whether mCAT might prevent the development of the SASP. We therefore assayed the conditioned media from treated cells for IL-6 secretion by ELISA (Figure 1H). We also assayed the mRNA levels of additional SASP factors in mtDNA-depleted cells (MMP3, IL1A, IL1B, IL6, and IL8) by quantitative PCR (Figure 1I). In both assays, *mCAT* did not suppress any SASP factors. Indeed, we observed a small—but not statistically significant—increase in some SASP factors in *mCAT*-treated cells relative to vectors treated in non-senescent WT cells. Overall, we observed no evidence for SASP suppression by *mCAT*. 

Mitochondrial ROS have also been implicated in driving nuclear DNA damage and DNA damage signaling during senescence [24,25,26]. We therefore stained treated cells for 53BP1 foci by immunofluorescence and quantified foci numbers per cell. Mitochondrial catalase had no discernable effect on the distribution of DNA damage foci numbers (Figure 1J,K) or mean numbers per experiment (Figure 1L). Thus, *mCAT* did not change the markers of DNA double-strand breaks. Together, our data indicated that *mCAT* does not significantly alter any markers of senescence following genotoxic or mitochondrial stress.

### 3.2. Gonadal Fat from Long-Lived mCAT Mice Accumulates Senescent Cells Normally

Senescent cells accumulate during aging, and eliminating them extends both life span and health span [18,22]. Since the origin of senescent cells during aging in vivo is not known and *mCAT* mice display extended lifespans [13], we analyzed aged WT and *mCAT* mice to determine if these mice were protected from senescent cell development. Gonadal adipose tissue was chosen for analysis, as this tissue displays an age-dependent accumulation of senescent cells—which are hypothesized to accumulate due to ROS [32]. This tissue also loses mtDNA copy numbers and gains heteroplasmy with age [33,34], develops senescent cells in response to mitochondrial dysfunction [23], shares a mesenchymal origin with fibroblasts, and can be relatively easily analyzed due to its semi-transparent nature [35]. Quantitative PCR analysis revealed statistically significant increases in both *p16^INK4a^* (Figure 2A) and *p21^WAF1^* (Figure 2B) at both 17 and 25–30 (25+) months of age relative to young (4 months) control mice; however, no differences were observed between WT and *mCAT* mice at either 17 or 25+ months. Additionally, we stained adipose tissue from aged WT and *mCAT* mice for senescence-associated beta-galactosidase (Figure 2C,D). No differences in beta-galactosidase intensity were observed at 25+ months (Figure 2C) or cell numbers at 17 months (Figure 2D). Furthermore, the loss of nuclear HMGB1, a marker of senescence [36], was likewise equivalent between WT and mCAT adipose tissues at 17 months of age (Figure 2E). Thus, no markers of senescent cell accumulation were changed by *mCAT* during chronological aging.

### 3.3. Gonadal Fat from Aged mCAT Mice Is Not Protected from Elevation of SASP Factors

While we did not observe changes in senescent cell accumulation in mCAT mice, we also investigated whether mCAT might act to limit the SASP in aged mice. For our analysis, we chose a panel of senescence-associated inflammatory cytokines (*Il6*, *Il1b*, *Il10*, *Cxcl1*, *Ccl2*, *Ccl11*, and *Tnf*—Figure 3A–G), tissue remodeling factors (*Mmp3*, *Plau*, *Serpine1*, and *Thbs2—*Figure 3H–K), and oxylipin synthases (*Alox5*, *Ptges*, and *Ptgs2*—Figure 3L–N). While most SASP factors were increased during aging in adipose tissue, a few were not (*Serpine1*, *Thbs2*, and *Ptgs2*). Additionally, a few factors were increased in 25+-month-old mice relative to 17-month-old mice (e.g., *Il10* and *Plau*); however, mCAT did not lower the expression of any SASP factors relative to the age-matched controls. Indeed, in the case of *Ptges*, the expression was increased in age-matched 17-month-old *mCAT* mice relative to WT mice (Figure 3M). Therefore, mCAT does not lower the expression of SASP factors during natural aging. Together with the lack of changes observed in the accumulation of senescent cells with mCAT (Figure 2), we conclude that mCAT does not influence the development of age-related senescence phenotypes in adipose tissue.

## 4. Discussion

Previous studies have implicated mitochondria and other ROS sources as a potential driver of both senescence and aging phenotypes [37,38,39]. While our study has some limitations, as noted below, it presents an interesting question: what if mitochondrial ROS are not responsible for the senescence phenotypes that occur during chronological aging? One possibility is that other sources of ROS or other reactive molecules could still drive senescence and the SASP. For example, NAD(P)H oxidases [40,41], lipoxygenases [42], and increased labile iron [43] have been shown to drive senescence. While the loss of many antioxidant enzymes can drive senescence and accelerate aging in mice [4,5,6,7], increasing levels of antioxidant enzymes has been less successful in antagonizing aging [11,12], suggesting that antioxidant enzyme levels are already sufficient to prevent naturally occurring ROS from driving aging in mice and presumably most mammals. The notable exception to this comes from the *mCAT* mouse, which lives longer than wild-type mice [13]; however, we show here that these mice are not protected from the development of senescent cells, at least in adipose tissue.

Alternatively, mitochondria may play a role in senescence and the SASP independent of ROS generation. We previously showed that multiple drivers of mitochondrial dysfunction can drive cellular senescence, but also limit the development of multiple aspects of the SASP [23]. Similar limitations appear when mitochondria are removed by enhanced mitophagy [24]. Mitochondria are required for the cellular oxidation of NADH, and if the cell is unable to compensate, it can undergo senescence [23]. Additionally, mitochondria can be sources of cytosolic DNA, which can activate interferon signaling via the cGAS-STING pathway [44]. This appears to occur in response to sub-apoptotic stress, and it may be independent of mitochondrial ROS. As such, it might be the case that mitochondrial DNA, rather than ROS, is a more important effector of the SASP. Indeed, in progeroid mitochondrial DNA mutator (*POLG^D257A^*) mice, which also accumulate senescent cells, the ablation of the CGAS-STING pathway extends their life span and protects against degenerative pathologies [45]. Thus, mitochondria may have additional progeronic roles in aging and senescence outside of ROS production.

It remains possible, if not likely, that the life span extension observed in *mCAT* mice is due to the preservation of function in tissues other than fat, likely those that are mitochondrially enriched, such as neurons or skeletal and cardiac muscle. Many of the key cell types in these tissues are protected by *mCAT* [46,47,48]. These cell types also tend to be postmitotic, so it is possible that they are less prone to the development of senescence.

This study carries some limitations. For example, we only analyzed aged gonadal adipose tissues in our animal studies. Thus, it is unclear if the observed phenotypes are broadly applicable to every *mCAT* mouse tissue. Nevertheless, it does appear to be the case that mitochondrial hydrogen peroxide is not a driving factor for senescence or the SASP during aging in this tissue and thus is unlikely to be a universal driver of senescent phenotypes. Notably, mCAT lowered levels of MitoSOX and DCFDA fluorescence in senescent cells (Figure 1A,B). This may initially appear surprising, as the catalase reaction that targets hydrogen peroxide and MitoSOX functions more as a sensor of mitochondrial superoxide; however, reports have since identified the detection of other radicals by MitoSOX and its parent molecule, dihydroethidium [49,50]. Other reports have similarly observed reductions in MitoSOX fluorescence in response to mCAT [51,52,53], which is consistent with our results. Conversely, DCFDA does not measure mitochondria-specific hydrogen peroxide and thus could suggest other sources of ROS. Therefore, there are technical limitations to these probes, even when fluorescence is reduced by mCAT.

Despite these caveats, our data indicate that mitochondrial ROS are unlikely to be a universal driver of age-related senescence or the SASP. This aligns with a broader array of literature challenging the free radical theory of aging, advocating instead for a more nuanced perspective. In this view, free radicals and oxidative stress are recognized as just one among several potential contributors to aging [11,54,55]. This interpretation is consistent with the observed senescent phenotypes and suggests that a similar view could be applied to senescence.

## Figures and Tables

**Figure 1 biomedicines-12-00414-f001:**
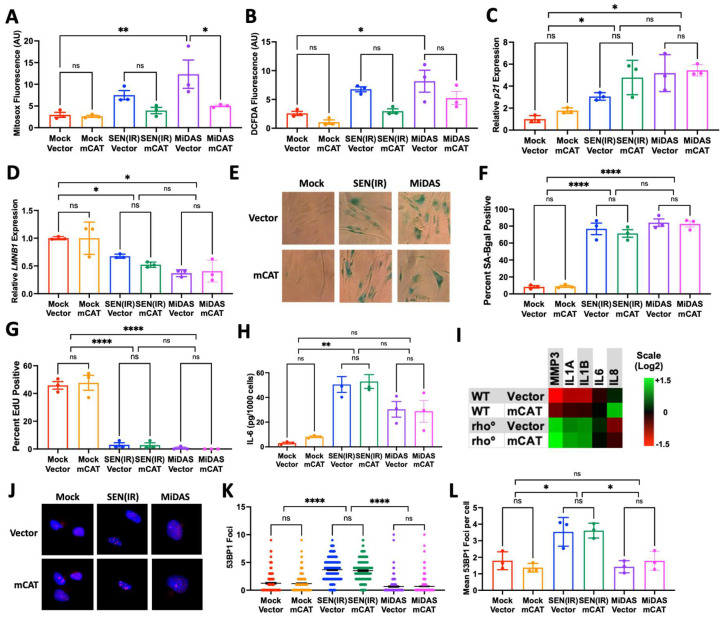
Mitochondrial catalase does not prevent senescence phenotypes in cultured cells. (**A**–**G**). IMR-90 fibroblasts were transduced with lentiviruses containing either an empty lentiviral vector or the mCAT insert and selected in puromycin for 2 days. Cells were then either irradiated with 10 Gy of ionizing radiation [SEN(IR)] and cultured for 10 days, depleted of mitochondrial DNA by continuous culture in EtBr for 2 weeks (MiDAS), or mock-irradiated and cultured continuously in growth media for 2 weeks (Mock). Cells were then analyzed for (**A**) MitoSOX fluorescence, (**B**) DCFDA fluorescence, (**C**) senescence-associated beta-galactosidase, (**D**) P21^WAF1^ mRNA expression, (**G**) 24 h EdU incorporation indices, or (**H**) IL-6 secretion. (**I**) Rho cells were transduced with mCAT or empty vector and cultured in the presence of pyruvate before RNA extraction and analysis for the indicated SASP factors. (**J**–**L**). Cells from (**A**–**H**) were analyzed by immunofluorescence for 53BP1 DNA damage foci. (**J**). Example of staining for 53BP1 foci. (**K**). Number of foci per cell, for 200 cells. (**L**). Mean 53BP1 DNA damage foci per replicate. All RNA levels were normalized to beta-actin. ns = not significant, * = *p* < 0.05, ** = *p* < 0.01, **** = *p* < 0.0001 for all panels (Brown–Forsythe ANOVA with Welch’s correction).

**Figure 2 biomedicines-12-00414-f002:**
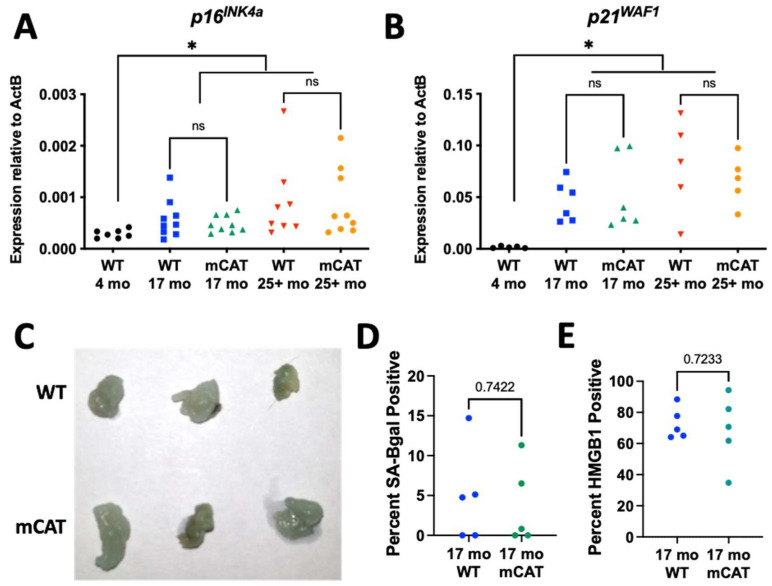
Long-lived mCAT mice are not protected from senescent cell accumulation. Gonadal adipose tissue from WT or *mCAT* mice at the indicated ages were analyzed for the following: (**A**) RNA levels of *p16^INK4a^* normalized to beta-actin and (**B**) RNA levels of *p21^WAF1^* normalized to beta-actin. (**C**) Representative images of fat stained for senescence-associated beta-galactosidase. (**D**) Tissues from (**C**) were sectioned, and beta-galactosidase-positive cells were counted and expressed as a percent of total nuclei detected. (**E**) Tissues from (**C**) were sectioned and Hmgb1-positive cells were counted and expressed as percent of total nuclei detected. ns = not significant, * = *p* < 0.05 for gene expression (Brown–Forsythe ANOVA with Welch’s correction). Unpaired two-tailed *t*-tests with Welch’s correction were used for (**D**,**E**).

**Figure 3 biomedicines-12-00414-f003:**
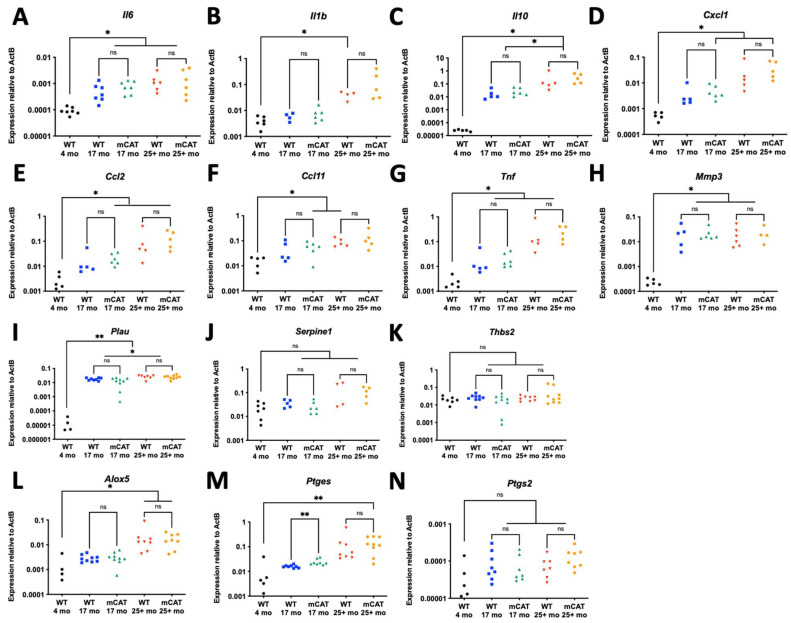
Mitochondrially targeted catalase does not affect the SASP in the fat of aged mice. (**A**–**N**) Gonadal adipose tissue from WT or mCAT mice at the indicated ages were analyzed for the following RNAs, and all RNAs normalized to beta-actin: (**A**) *Il6*, (**B**) *Il1b*, (**C**) *Il10*, (**D**) *Cxcl1*, (**E**) *Ccl2*, (**F**) *Ccl11*, (**G**) *Tnf*, (**H**) *Mmp3*, (**I**) *Plau*, (**J**) *Serpine1*, (**K**) *Thbs2*, (**L**) *Alox5*, (**M**) *Ptges*, (**N**) *Ptgs2*. ns = not significant, * = *p* < 0.05, ** = *p* < 0.01 for all panels (Brown–Forsythe ANOVA with Welch’s correction).

## Data Availability

All data are presented in the manuscript.

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
