# Peer review of "Mitochondria-Targeted Catalase Does Not Suppress Development of Cellular Senescence during Aging"

_biomedicines, 2024, doi:10.3390/biomedicines12020414_

Round 1

Reviewer 1 Report

Comments and Suggestions for Authors

The paper “Mitochondria-targeted catalase does not suppress development of cellular senescence during aging” investigated the impact of mCAT on senescence in cultured cells and aged mice, expecting to identify whether mCAT or the SASP could prevent senescence. The article conforms to readers' interest of biomedicines with sufficient references. However, there are some shortcomings that need to be further improved or explained.

Comments:

Q1. The description of the background in the abstract is too long, which does not reflect the methods and results of this paper well.

Q2. Line 40-45, please confirm whether these expressions are contradictory with the tittle.

Q3. In the last paragraph of introduction section (Line 60-68), had the relevant results been worked out in reference 13? Please supplemented the research contents in this paper, as well as the research significance.

Q4. Materials and Methods, please supplement materials information and subtitles of these methods.

Q5. Please confirm the presentation of *= p<0.05?

Q6. It is suggested to unify the presentation format of the chart in this paper, bar chart or scatter chart.

Q7. This paper is very innovative, and because of this, only the high and low expression of a few proteins at the cellular level is not enough to support the conclusion of this paper.

Q8. What are the evaluation criteria of cellular senescence, and how to determine the success of cell modeling in this paper?

Q9. This article has been referring to the relationship between cellular senescence and longevity. Is there any direct evidence that the cell senescence induced by this model is related to the corresponding animal life span?

Q10. It is suggested to add a schematic diagram to concatenate the relationship between determined protein/gene expressions and aging in this paper. In addition, this paper lacks valid conclusions, which should not be comprised of any references.

Comments on the Quality of English Language

Moderate editing of English language required

Reviewer 2 Report

Comments and Suggestions for Authors

The authors presented a very interesting study aimed at shedding light on the mechanisms of aging and the role of oxidative stress and mitochondrial reactive oxygen species in this process. Two models were used - cellular senescence and old mice. In both cases, the authors induced senescence or aging and analyzed the phenotype associated with aging and the influence of mitochondrial catalase expression on it. Unexpectedly, the authors found no significant effects of mitoCAT on the aging phenotype. This is particularly curious since mitoCAT transgenic mice, as well as similar cells, were previously described by Rabinovitch (who provided them to the authors) as significantly reversing aging and extending lifespan by reducing aging markers.

I believe that any negative data should be published, even if it refutes the original theory. Otherwise, a traditional publication bias occurs and we place false hopes on various promising approaches and only see positive results such as mito-catalase.

However, I have a number of questions about the study design and presentation of the results that need to be addressed before the manuscript can be considered for publication.

1. In the Methods section, provide more detailed information about the age of the mice, including the correlation between the age of the wild-type and transgenic mice. Different ages are mentioned in the Results section, but it is unclear how these were determined.

2. IMR-90 cells are lung fibroblasts, why were they chosen as a model? Meanwhile, analysis in mice focused on adipose tissue.

3. Why was gamma-irradiation used as a model? It is not the best model for aging, but rather a model for radiation damage.

4. The same question applies to the Rho-null cell model. Why do the authors consider it a model for aging? Several experiments have shown that the number of mitochondria can actually increase with aging in some tissues. In addition, how do the authors envision MitoCAT acting in cells where mitochondria are absent or damaged? MitoCAT should be targeted to the mitochondria. Furthermore, the authors did not prove that they actually obtained Rho-null cells. For example, they could have checked the amount of mtDNA or the number of mitochondria or the mitochondrial potential.

5. In which passage were the fibroblasts? With IMR-90 it is known that replicative senescence occurs at high passages.

6. Why was mitoSOX used instead of probes for hydrogen peroxide? For example Hyper protein vector or at least DCF or CellROX.

7. Why was fat chosen for analysis in transgenic and normal aging mice?

8. The quantitative data presented in the article should be accompanied by representative images to accurately verify that the desired parameter was measured. For example, images of cells stained with MitoSOX should be presented to show that ROS generation occurs specifically in these cells' mitochondria. For fat histology, photos of tissues stained with beta-galactosidase should be included. The same applies to immunostaining for 53BP1 and HMGB1. These images can be added to the main text or as supplementary material. Conversely, macroscopic images of fat preparations are not informative and can be removed.

Comments on the Quality of English Language

The English language is satisfactory, but a typo check and formatting correction are required. Follow the MDPI template

Round 2

Reviewer 1 Report

Comments and Suggestions for Authors

No additional problems.

Author Response

Thank you

Reviewer 2 Report

Comments and Suggestions for Authors

The authors have significantly improved their manuscript by trying to answer my comments and questions. I only have one more question about the measurement of reactive oxygen species. The authors present numerical data of image analysis after loading the cells with fluorescent probes for ROS. At the same time, the authors claim that they do not have representative images of these cells because the researcher who performed these experiments passed away. This is very strange. How are calculations of fluorescence intensity done if there are no images? I have serious doubts that this data can be trusted. If the authors have not analyzed the images themselves, how can they be sure that the data presented is correct?

Comments on the Quality of English Language

No comments

Author Response

We apologize for the confusion.  The analysis was performed using flow cytometry, not captured images, as mentioned in the Materials and Methods section - so quantitation was never performed on captured images.  Images were only captured to confirm localization of the probes and were not thresholded for quantitative assessment, so they would not be useful for this purpose, even if they were available.  However, we understand the need for validation of the probe usage.  We have therefore added histograms from our flow cytometry (FCS) files plotting cell event counts against MitoSOX or DCFDA fluorescence.

To clarify, Judith Campisi did not perform these analyses.  Dr. Wiley performed the analysis while training under Dr. Campisi at the Buck Institute several years ago, but Dr. Wiley is no longer at the Buck Institute (where these analyses were performed) and Dr. Campisi is deceased.  Therefore access to the image files, not the investigator that performed the analysis, is the concern.  As mentioned above, the images were simply captured in a flask to confirm localization - not on an optical grade surface that can be used for quantitation. 

Round 3

Reviewer 2 Report

Comments and Suggestions for Authors

I have no additional comments.

Comments on the Quality of English Language

Quality of English is sufficient.